# Comparative Analysis of the Frequency of Fresh and Frozen Fish Consumption Among Two Cohorts of Pregnant Women

**DOI:** 10.3390/nu17030439

**Published:** 2025-01-25

**Authors:** Angela Alibrandi, Carlo Giannetto, Agata Zirilli, Vihra Dimitrova, Giosuè Giordano Incognito, Roberta Granese, Angelina De Pascale, Maurizio Lanfranchi

**Affiliations:** 1Department of Economics, University of Messina, 98100 Messina, Italy; aalibrandi@unime.it (A.A.); azirilli@unime.it (A.Z.); angela.depascale@unime.it (A.D.P.); maurizio.lanfranchi@unime.it (M.L.); 2Department of Management, University of Agribusiness and Rural Development, BG-4003 Plovdiv, Bulgaria; vihra.dimitrova@abv.bg; 3Department of General Surgery and Medical Surgical Specialties, University of Catania, 95123 Catania, Italy; giosuegiordano.incognito@studium.unict.it; 4Department of Biomedical and Dental Sciences and Morphofunctional Imaging, “G. Martino” University Hospital, 98100 Messina, Italy; roberta.granese@unime.it

**Keywords:** health in pregnancy, fresh and frozen fish consumption, statistical comparisons, consumer behavior analysis

## Abstract

Objectives: today’s consumers are increasingly interested in the relationship between food and health, recognizing food as a means to meet nutritional needs and prevent diseases. A diet rich in fish is beneficial to health, potentially protecting against cancer, cardiovascular, autoimmune, neurodegenerative, and metabolic diseases. During pregnancy, adequate nutrition benefits both the mother and the unborn child. This study compares pregnant women from a decade ago with those recently enrolled to evaluate differences in eating styles, specifically the consumption of fresh and frozen fish. Methods: we compared 114 pregnant women from 2013 with 168 women from 2023, using the same questionnaire to evaluate their eating habits during pregnancy, focusing on fresh and frozen fish consumption. Variables for statistical analyses included age, education, profession, family size, pre-pregnancy BMI, differential BMI, and frequency of fish consumption. Results: the comparison showed an increase in fish consumption, both fresh and frozen, among pregnant women in 2023 compared to 2013, indicating greater awareness of the health benefits of fish. Changes in dietary habits were influenced by profession, education level, and family size. Women in the 2023 cohort experienced smaller weight gain during pregnancy, suggesting potential health benefits. These shifts likely result from improved nutrition education and access to healthy foods, highlighting the importance of public health efforts to enhance maternal and fetal health. Conclusions: significant changes in the dietary habits of pregnant women over a decade were observed, with increased fish consumption in 2023 compared to 2013. These findings emphasize the role of nutrition education and improved access to healthy foods in promoting maternal and fetal health.

## 1. Introduction

Today’s consumers are increasingly interested in the relationship between food and health, as food serves not only as a means to meet the body’s nutritional needs but also as a tool for preventing diseases [1]. For example, fish has a simultaneous presence of vitamins (including vitamins A, B12, D, and E), proteins, and omega-3 [2]. The consumption of fish, due to its reduced caloric intake, is a significant benefit in terms of preventing overweight and obesity [3,4]. A diet rich in fish is beneficial to consumers’ health and may have a protective effect against certain types of cancer, cardiovascular, autoimmune, neurodegenerative, and ocular diseases [5,6,7,8,9,10,11,12]. Nutrients such as protein, long-chain omega-3 polyunsaturated fatty acids (LCPUFA), selenium, iodine, and vitamin D, which are abundant in fish, are also considered favorable for the course of pregnancy, as well as for fetal growth and development [13,14].

The Italian coastline stretches for 9136 km, accounting for 8.75% of the EU coastline. While the fishing industry contributes just over 0.5% to the national GDP, its impact is more pronounced in specific areas, particularly in the southern regions [15]. As for the per capita fish consumption in Italy from 2013 to 2023, the FAO provides detailed data through its FAOSTAT platform. These data show a steady growth in fish consumption in Italy, indicating an increase of about 7 kg per capita over a decade. This consumption had been steadily increasing for about two decades at a rate of +2% per year. The rise in consumption of canned fish, canned tuna, and frozen foods played a significant role in this trend.

In addition, the fishing sector in Italy is expected to benefit from the increased demand for sustainable and locally sourced seafood products, particularly as consumers become more concerned with environmental issues and sustainability [16,17,18]. The growing interest in environmentally friendly and organic food options, combined with the rise of e-commerce in food sales, is driving the development of new markets and distribution channels. The Italian fishing industry will also be able, through investments in sustainability and innovative packaging solutions, to comply with such trends. Such evolution in consumer behavior should also allow for the strengthening of local fisheries and contribute to the economic growth of regions where such activities are of importance [19,20].

Given these considerations, this study aims to compare a cohort of pregnant women enrolled a decade ago with women enrolled recently (one year ago) to explore two primary research questions: (1) Have there been significant changes in the frequency and type of fish consumption among pregnant women in the past decade? (2) How do demographic and socioeconomic factors influence these dietary patterns? By addressing these questions, the study aims to elucidate the evolving dietary patterns of pregnant women and provide actionable insights to improve maternal and fetal health outcomes. These findings will inform public health strategies to enhance nutritional education and access to healthy foods.

## 2. Materials and Methods

Specifically, we used the cohort of 114 pregnant women enrolled in 2013 [21] and the cohort of 248 women enrolled in 2023 [22] who filled out the same questionnaire administered to evaluate their eating habits during pregnancy. The questionnaire was administered at the time of the women’s discharge [23,24]. The women of both cohorts provided their informed consent to the investigation in the protection of their privacy and the guarantee of anonymity, subject to the favorable opinion of the ethical committee. Since the first cohort is composed of women who consume both fresh and frozen fish, women who do not consume fish (18) and women who only consume fresh fish (62) were excluded from the 2023 cohort. The reason the two cohorts were chosen is due to the fact that the two cohorts of women agreed to fill out the same questionnaire. Therefore, the same variables were detected in the same form in two different periods. In doing so, it was possible to compare the two cohorts, the first made up of 114 units and the second of 168.

We are aware that the research design has a limit due to the self-selection bias: in fact, each pregnant woman has the possibility to choose whether to adhere (or not) to the compilation of the questionnaire and, therefore, to become part of the sample subject to statistical analysis. A pregnant woman who, for various reasons, does not provide consent to participate in the survey represents an impoverishment of available information, and consequently, this involves a self-selection of the interviewees, which cannot be controlled by the researcher.

The sampling chosen guarantees representativeness as it allows the reaching of a small “photograph” of the pregnant female population that consumes fish; the chosen sample allows the conclusion that can be generalized to the reference population, thanks also to the use of inferential procedures.

The variables considered for statistical analyses were the following: age, education, profession, family size (small: ≤3 members; large: >3 members), pre-pregnancy BMI, and differential BMI (considered to be the difference between the BMI at the end of pregnancy and the pre-pregnancy BMI), frequency of consumption of fresh and frozen fish (monthly or less, fortnightly, weekly, biweekly or more). These variables were detected in the two cohorts and compared with each other. The evaluation of each analyzed variable assumes a great influence: age, education level, occupation, and family size represent the stratification variables; the pre-pregnancy BMI and differential BMI instead constitute indicators of the physical health status of the mothers.

The numerical variables were expressed as mean and standard deviation (SD), and the categorical variables as absolute frequencies and percentages. In order to compare the two cohorts of pregnant women, the comparison between proportions was applied according to the frequency of fresh and frozen fish consumption, taking into account the profession, the educational level, and the family size [25]. Student *t*-test was applied in order to compare the two cohorts of pregnant women with reference to age, pre-pregnancy BMI, and differential BMI. A boxplot was realized to better visualize the distribution of differential BMI into two cohorts. A *p*-value lower than 0.05 was considered statistically significant and reported in bold. All statistical analyses were performed using the SPSS package, version 22.0.

## 3. Results

The ages of the women belonging to the two cohorts do not differ significantly (1 cohort 32.39 ± 5.13; 2 cohort 31.12 ± 5.64; *p* = 0.055).

Table 1 shows crosstabulation referred to the comparison between the two cohorts according to the frequency of fresh and frozen fish consumption.

As can be seen from the results reported in Table 1, we note a propensity on the part of the pregnant women belonging to the second cohort towards a more frequent consumption of fish, both fresh and frozen. More specifically, for fresh fish, there was a significant reduction in fortnightly consumption (*p* = 0.002), which is accompanied by a significant increase in biweekly consumption (*p* = 0.046); for frozen fish, a significant reduction is found for monthly consumption (*p* = 0.001), while significant increases are found for weekly consumption (*p* = 0.004) and biweekly consumption (*p* = 0.015).

In Table 2, the fish consumption of pregnant women of the two cohorts was analyzed stratifying for different types of professions.

As regards housewives, the consumption of fresh fish between the two cohorts does not undergo significant variations; as regards, however, the consumption of frozen fish, a significant decrease is evident in monthly consumption (*p* = 0.009) while a significant increase is evident in weekly consumption (*p* = 0.010).

As regards pregnant managers, the women of the second cohort significantly decreased their monthly and fortnightly consumption of fresh fish (*p* = 0.001 and *p* < 0.001, respectively) while increasing their weekly consumption (*p* < 0.001). As regards frozen fish, however, we find a significant reduction in fortnightly consumption (*p* = 0.030), which is accompanied by an increase in biweekly consumption (*p* = 0.027).

As regards unemployed/students in the second cohort and in relation to fortnightly consumption, for fresh fish, we note a significant reduction (*p* < 0.001) and, at the same time, a significant increase for frozen fish (*p* < 0.001). In the same category of pregnant women, a significant increase in weekly consumption of fresh fish was noted (*p* = 0.002).

Examining the last stratum, i.e., pregnant workers/employees, we note how in the second cohort compared to the first there is a significant decrease in the monthly consumption of fresh and frozen fish (*p* = 0.014 and *p* < 0.001, respectively) and in the fortnightly consumption of fresh fish (*p* = 0.003); a significant increase, however, occurs for the weekly consumption of frozen fish (*p* < 0.001) and the biweekly consumption of both types of fish (*p* = 0.001).

As regards educational qualifications (Table 3), it is noted that the fortnightly consumption of fresh fish undergoes a significant decrease both in the subpopulation of women with elementary/middle school (*p* = 0.001), in women with a diploma (*p* = 0.002) and with a degree (*p* = 0.002). In addition, we can note that the biweekly consumption of fresh fish undergoes a significant increase both in the subpopulation of women with elementary/middle school (*p* = 0.001) and with a diploma (*p* = 0.017); in fact, even for women with a degree there is an increase, but this is not statistically significant (*p* = 0.532). Within this last group, the monthly consumption of fresh fish presents a significant decrease (*p* = 0.002), which contrasts with a significant increase in weekly consumption (*p* = 0.004).

In relation to frozen fish, the three subpopulations of women defined based on educational qualifications present similar behavior. In fact, monthly consumption undergoes a significant decrease (*p* = 0.018, *p* = 0.019, and *p* < 0.001, respectively), an increase in weekly consumption significant only for women with elementary/middle school education (*p* < 0.001) and for women with university degrees (*p* = 0.002). In the comparison between the two cohorts related to women in elementary/middle school, we note for the first time a significant decrease in the biweekly consumption of frozen fish (*p* = 0.016), but these data are to be considered difficult to extend, as they are linked to a small number of subjects into the compared groups (2 vs). For “diploma” and “degree” qualifications, the biweekly consumption of frozen fish shows a significant increase (*p* = 0.009 and *p* = <0.001, respectively).

Examining the results shown in Table 4, in which we stratified for small families (≤3 components) and large families (>3 components), we note that the fortnightly consumption of fresh fish within small families undergoes a significant reduction (*p* = 0.006), while the biweekly consumption significantly increases (*p* = 0.018). In large families, there is a significant decrease in monthly (*p* < 0.001) and fortnightly (*p* < 0.001) consumption, while weekly consumption of fresh fish significantly increases (*p* < 0.001).

In reference to frozen fish consumption, small families and large families present similar behavior in the comparison between the two cohorts: monthly consumption shows a highly significant reduction (*p* < 0.001 for small and large families), while we note a significant increase for weekly consumption (*p* = 0.001 for small families and *p* < 0.001 for large families) and for biweekly consumption (*p* = 0.027 for small families and *p* = 0.002 for large families).

Finally, we focused on the BMI of pregnant women, comparing the two cohorts, both in reference to the initial (pre-pregnancy) BMI and the differential BMI (Table 5).

Examining the results of the comparison between pre-pregnancy BMI, we can see that there are no significant differences between the two cohorts (*p* = 0.408). On the contrary, the differential BMI is significantly higher in cohort 1 than in cohort 2 (*p* = 0.022), and this result highlights that women in the second cohort experience a smaller increase in weight during pregnancy. This comparison can be better visualized in the boxplot shown in Figure 1.

## 4. Discussion

The comparison of the two cohorts of pregnant women from 2013 and 2023 reveals significant changes in their fish consumption habits. The findings highlight a notable shift in dietary preferences over the analyzed decade. The analysis shows that pregnant women in 2023 are more inclined to consume fish, both fresh and frozen than their counterparts in 2013. This shift suggests a trend towards more frequent inclusion of fresh fish in their diets. Meanwhile, regarding the consumption of frozen fish, the changes are even more pronounced. Indeed, on the one hand, there emerges a significant reduction in monthly consumption; on the other hand, there is a significant increase both in weekly and biweekly consumption.

These results imply that pregnant women in 2023 are not only incorporating fish into their diets more frequently but are also diversifying the types of fish they consume. The increased frequency of fish consumption, particularly on a weekly and biweekly basis, reflects an overall positive trend toward integrating nutrient-rich foods into the maternal diet.

The observed differences are likely due to changes in dietary preferences and behaviors rather than demographic shifts [26,27,28]. The stratified analysis of fish consumption by profession among pregnant women in the 2013 and 2023 cohorts provides deeper insights into how dietary habits have evolved across different professional groups. The findings indicate distinct trends based on the type of profession. For housewives, the consumption of fresh fish did not show significant changes between the two cohorts, indicating stability in this group’s dietary habits regarding fresh fish. However, the consumption of frozen fish showed a significant shift. Indeed, this research showed that there was a decrease in monthly consumption and a significant increase in weekly consumption. Pregnant managers exhibited significant changes in their fish consumption patterns. It is observed that there was a notable decrease in both monthly and fortnightly consumption of fresh fish while the weekly consumption significantly increased. This shift indicates a move towards more frequent but smaller portions of fresh fish. For frozen fish, there was a reduction in fortnightly consumption and an increase in biweekly consumption, reflecting a similar trend of more regular but spaced-out consumption. Among unemployed women and students, significant changes were observed as well. There was a significant reduction in fortnightly consumption of fresh fish and a corresponding significant increase in fortnightly consumption of frozen fish. Additionally, there was a significant increase in the weekly consumption of fresh fish. These patterns suggest a diversification in fish consumption, with a notable shift towards more frequent inclusion of both fresh and frozen fish. For pregnant workers and employees, the results show a significant decrease in monthly consumption of both fresh and frozen fish and fortnightly consumption of fresh fish. However, there was a significant increase in weekly consumption of frozen fish and biweekly consumption of both types of fish. These variations underscore the impact of socioeconomic factors on dietary habits [29].

This indicates a shift towards more frequent consumption of fish, particularly frozen fish, reflecting perhaps greater convenience and storage benefits associated with frozen products [30]. The variations in fish consumption habits across different professional groups highlight the impact of lifestyle, convenience, and economic factors on dietary choices [31]. The significant increases in more frequent consumption of both fresh and frozen fish across most professional categories suggest an overall positive trend towards incorporating fish into regular diets [32]. This is likely influenced by increased awareness of the health benefits of fish consumption during pregnancy, improvements in food safety, and better access to a variety of fish options. The stratified analysis based on educational qualifications offers additional insights into the changes in fish consumption habits among pregnant women from the two cohorts, emphasizing how educational background influences dietary behavior.

These results may reflect increased awareness and knowledge of the health benefits of regular fish consumption during pregnancy, possibly driven by better access to nutritional information and resources [33]. The results stratified by family size and BMI provide further insights into the fish consumption patterns among pregnant women from the 2013 and 2023 cohorts. These findings highlight how family structure and BMI influence dietary habits. When comparing the two cohorts based on BMI, both initial (pre-pregnancy) BMI and differential BMI were considered (Table 5). Although the specific results for BMI are not detailed here, the analysis of BMI is crucial as it provides context for understanding the nutritional status and health outcomes of pregnant women in relation to their dietary habits.

The findings indicate a clear trend towards more frequent consumption of both fresh and frozen fish across different family sizes. This shift reflects increased awareness of the nutritional benefits of fish, as well as improved access to a variety of fish options. The changes in consumption patterns are consistent across different family sizes.

The findings regarding family size and BMI further illustrate how dietary habits are influenced by household dynamics and individual health status. The more frequent fish consumption across different family sizes highlights an overall improvement in dietary practices, potentially driven by increased awareness of family nutrition needs. This trend is consistent with findings from Bangladesh, where interventions combining agricultural training and nutrition behavior change communication led to an increase in the production and consumption of nutrient-dense foods, including fish, particularly in households with diversified homestead food production. These findings highlight the importance of integrating nutrition education with household-level food production strategies to promote dietary improvements [34]. Similarly, research on Israeli households demonstrates how socioeconomic and demographic factors, such as household size and income, influence food purchasing behaviors. Larger households often achieve economies of scale in food consumption, while higher-income families allocate more resources to a greater variety of nutrient-dense foods, including fish [35]. Together, these studies provide evidence that the observed trend of improved dietary practices across family sizes is consistent with broader research linking household characteristics, socioeconomic status, and targeted interventions to positive dietary outcomes.

The comparison of pre-pregnancy BMI between the two cohorts reveals no significant differences, indicating that the baseline nutritional status and body composition of the women in the 2013 and 2023 cohorts were similar.

This study has several strengths. Among them, the study utilized identical questionnaires for both cohorts, enabling direct and reliable comparisons across time. The inclusion of diverse demographic and socioeconomic strata further enhances the generalizability of the findings, providing a comprehensive view of changes in dietary patterns. However, this study also has limitations that must be acknowledged. First, the reliance on self-reported data introduces the potential for recall bias, as participants may not accurately remember their dietary habits. Second, the study design was observational and cross-sectional, which limits the ability to infer causal relationships between fish consumption and the observed changes in health outcomes. Finally, while the study stratified data by several factors, unmeasured variables could also have influenced the findings. Future studies addressing these limitations, including longitudinal designs and objective measures of dietary intake, could provide deeper insights into the long-term implications of fish consumption during pregnancy for maternal and neonatal health.

## 5. Conclusions

This study has highlighted significant changes in the dietary habits of pregnant women over a decade. The results show an increase in the frequency of fish consumption, both fresh and frozen, among pregnant women in 2023 compared to those in 2013. This trend reflects greater awareness of the nutritional benefits associated with fish consumption during pregnancy, underscoring an increased focus on maternal and fetal health. Stratified analyses by profession, education level, and family size further emphasized how these factors influence dietary choices, with significant variations observed across different socioeconomic groups from the two cohorts.

The increase in weekly and biweekly fish consumption indicates a positive shift towards a more balanced diet rich in essential nutrients. Moreover, the data reveal that women in the second cohort experienced a smaller increase in weight during pregnancy, suggesting that these dietary changes could have contributed to better weight management.

These findings suggest that nutrition education initiatives and policies to improve access to healthy foods may have played a crucial role in promoting better dietary practices among pregnant women. This underscores the importance of continued public health efforts and targeted interventions to support nutritional education and access to healthy foods.

Future research could further explore the long-term impacts of these dietary habits on maternal and neonatal health, contributing to the development of effective strategies to enhance nutrition during pregnancy. Such studies could provide deeper insights into how sustained dietary improvements can affect health outcomes for both mothers and their children, ultimately supporting the creation of comprehensive nutritional guidelines and policies for pregnant women.

In future perspectives, longitudinal cohorts could be started to investigate other possible variables related to fish consumption during pregnancy and the possible benefits deriving from it.

## Figures and Tables

**Figure 1 nutrients-17-00439-f001:**
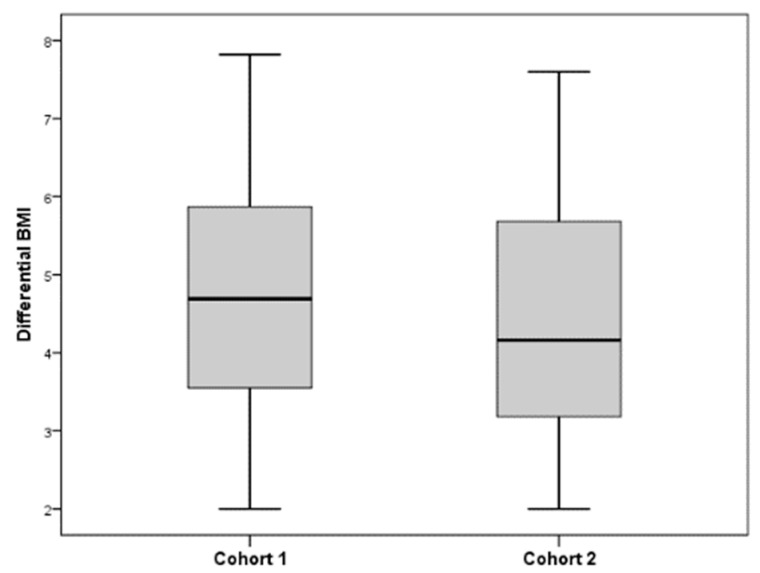
Boxplot of two cohorts related to differential BMI.

**Table 1 nutrients-17-00439-t001:** Comparison between cohorts according to the frequency of fresh and frozen fish consumption.

Frequency		FRESH FISH			FROZEN FISH		
of Consumption		Cohort 1	Cohort 2	*p*-Value	Cohort 1	Cohort 2	*p*-Value
Monthly	Count	19	22	0.401	77	78	0.001
	%	16.7%	13.1%		67.5%	46.4%	
Fortnightly	Count	19	9	0.002	12	13	0.416
	%	16.7%	5.4%		10.5%	7.7%	
Weekly	Count	43	69	0.567	20	55	0.004
	%	37.7%	41.1%		17.5%	32.7%	
Biweekly	Count	33	68	0.046	5	22	0.015
	%	28.9%	40.5%		4.4%	13.1%	

**Table 2 nutrients-17-00439-t002:** Comparison between cohorts according to frequency of fresh and frozen fish consumption and profession.

		FRESH FISH			FROZEN FISH		
**Housewife**		**Cohort 1**	**Cohort 2**	***p*-Value**	**Cohort 1**	**Cohort 2**	***p*-Value**
Monthly	Count	9	15	0.999	24	36	0.254
	%	25.0%	25.0%		66.7%	60.0%	
Fortnightly	Count	4	5	0.431	4	2	0.009
	%	11.1%	8.3%		11.1%	3.3%	
Weekly	Count	15	23	0.567	5	16	0.010
	%	41.7%	38.3%		13.9%	26.7%	
Biweekly	Count	8	17	0.251	3	6	0.630
	%	22.2%	28.3%		8.3%	10.0%	
**Manager**		**Cohort 1**	**Cohort 2**	***p*-Value**	**Cohort 1**	**Cohort 2**	***p*-Value**
Monthly	Count	3	1	0.001	7	10	0.289
	%	17.6%	4.8%		41.2%	47.6%	
Fortnightly	Count	3	0	<0.001	2	1	0.030
	%	17.6%	0.0%		11.8%	4.8%	
Weekly	Count	3	10	<0.001	7	7	0.177
	%	17.6%	47.6%		41.2%	33.3%	
Biweekly	Count	8	10	0.934	1	3	0.027
	%	47.1%	47.6%		5.9%	14.3%	
**Unemployed/student**		**Cohort 1**	**Cohort 2**	***p*-Value**	**Cohort 1**	**Cohort 2**	***p*-Value**
Monthly	Count	2	2	0.264	21	11	0.003
	%	7.7%	11.8%		80.8%	64.7%	
Fortnightly	Count	7	1	<0.001	2	5	<0.001
	%	26.9%	5.9%		7.7%	29.4%	
Weekly	Count	12	11	0.002	2	1	0.552
	%	46.2%	64.7%		7.7%	5.9%	
Biweekly	Count	5	3	0.733	1	0	0.011
	%	19.2%	17.6%		3.8%	0.0%	
**Worker/employee**		**Cohort 1**	**Cohort 2**	***p*-Value**	**Cohort 1**	**Cohort 2**	***p*-Value**
Monthly	Count	5	4	0.014	25	21	<0.001
	%	14.3%	5.7%		71.4%	30.0%	
Fortnightly	Count	5	3	0.003	4	5	0.213
	%	14.3%	4.3%		11.4%	7.1%	
Weekly	Count	13	25	0.812	6	31	<0.001
	%	37.1%	35.7%		17.1%	44.3%	
Biweekly	Count	12	38	0.001	0	13	<0.001
	%	34.3%	54.3%		0.0%	18.6%	

**Table 3 nutrients-17-00439-t003:** Comparison between cohorts according to the frequency of fresh and frozen fish consumption and educational level.

		FRESH FISH			FROZEN FISH		
**Elementary/Middle School**		**Cohort 1**	**Cohort 2**	***p*-Value**	**Cohort 1**	**Cohort 2**	***p*-Value**
Monthly	Count	4	9	0.323	17	25	0.018
	%	18.2%	23.1%		77.3%	64.1%	
Fortnightly	Count	6	4	0.001	2	3	0.676
	%	27.3%	10.3%		9.1%	7.7%	
Weekly	Count	10	15	0.242	1	10	<0.001
	%	45.5%	38.5%		4.5%	25.6%	
Biweekly	Count	2	11	0.001	2	1	0.016
	%	9.1%	28.2%		9.1%	2.6%	
**Diploma**		**Cohort 1**	**Cohort 2**	***p*-Value**	**Cohort 1**	**Cohort 2**	***p*-Value**
Monthly	Count	10	11	0.765	36	34	0.019
	%	18.9%	17.5%		67.9%	54.0%	
Fortnightly	Count	7	2	0.002	6	5	0.335
	%	13.2%	3.2%		11.3%	7.9%	
Weekly	Count	23	26	0.727	8	14	0.139
	%	43.4%	41.3%		15.1%	22.2%	
Biweekly	Count	13	24	0.017	3	10	0.009
	%	24.5%	38.1%		5.7%	15.9%	
**Degree**		**Cohort 1**	**Cohort 2**	***p*-Value**	**Cohort 1**	**Cohort 2**	***p*-Value**
Monthly	Count	5	2	0.002	24	19	<0.001
	%	12.8%	3.0%		61.5%	28.8%	
Fortnightly	Count	6	3	0.002	4	5	0.426
	%	15.4%	4.5%		10.33%	7.6%	
Weekly	Count	10	28	0.004	11	31	0.002
	%	25.6%	42.4%		28.2%	47.0%	
Biweekly	Count	18	33	0.532	0	11	<0.001
	%	46.2%	50.0%		0.0%	16.7%	

**Table 4 nutrients-17-00439-t004:** Comparison between cohorts according to frequency of fresh and frozen fish consumption and family size.

		FRESH FISH			FROZEN FISH		
**Small family**		**Cohort 1**	**Cohort 2**	***p*-Value**	**Cohort 1**	**Cohort 2**	***p*-Value**
Monthly	Count	18	16	0.595	74	47	<0.001
	%	16.2%	13.9%		66.7%	40.9%	
Fortnightly	Count	18	7	0.006	12	13	0.896
	%	16.2%	6.1%		10.8%	11.3%	
Weekly	Count	43	43	0.826	20	41	0.001
	%	38.7%	37.4%		18.0%	35.7%	
Biweekly	Count	32	49	0.018	5	14	0.027
	%	28.8%	42.6%		4.5%	12.2%	
**Large family**		**Cohort 1**	**Cohort 2**	***p*-Value**	**Cohort 1**	**Cohort 2**	***p*-Value**
Monthly	Count	1	6	<0.001	3	31	<0.001
	%	33.3%	11.3%		100.0%	58.5%	
Fortnightly	Count	1	2	<0.001	0	0	N.A.
	%	33.3%	3.8%		0.0%	0.0%	
Weekly	Count	0	26	<0.001	0	14	<0.001
	%	0.0%	49.1%		0.0%	26.4%	
Biweekly	Count	1	19	0.666	0	8	0.002
	%	33.3%	35.8%		0.0%	15.1%	

**Table 5 nutrients-17-00439-t005:** Comparison between cohorts (Mean ± SD and *p*-value) according to Pre-pregnancy BMI and Differential BMI.

BMI	Cohort 1	Cohort 2	*p*-Value
Pre-pregnancy	23.59 ± 4.19	24.05 ± 4.96	0.408
Differential	4.95 ± 1.79	4.31 ± 2.51	0.022

## Data Availability

The data presented in this study are available on request from the corresponding author due to privacy reasons.

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
