# Peer review of "Comparative Analysis of the Frequency of Fresh and Frozen Fish Consumption Among Two Cohorts of Pregnant Women"

_nutrients, 2025, doi:10.3390/nu17030439_

Round 1
Reviewer 1 Report
Comments and Suggestions for Authors
Introduction
Lines 41-42: ‘For example, fish has a simultaneous presence of vitamins, proteins, and omega-3.’
Comment: Clarify by mentioning specific examples of vitamins present in fish (for example, vitamin D or B12).
The introduction could be enriched by adding clear hypotheses or research questions to give more structure to the article and guide the reader to the precise aims of the study
Research design
Comments: it would be interesting to mention more details on potential limitations, such as the self-selection bias of respondents to the questionnaire.
Lines 97-99: ‘Specifically, we used the cohort of 114 pregnant women enrolled in 2013...’.
Comment: Explain briefly why these specific cohorts were chosen.
Methods
The methods are detailed but lack justification for certain approaches, such as excluding women who do not eat fish.
Comments: adding information on the validation of the questionnaire used and the reasons behind the sampling chosen would strengthen this section.
Lines 104-106: ‘Since the first cohort is composed of women who consume both fresh and frozen fish...’.
Comment: Justify why women who do not eat fish or who eat only fresh fish were excluded.
Lines 108-111: The methods section could be enriched by explaining why the following variables were considered in the statistical analysis: age, education level, occupation, family size, pre-pregnancy BMI, and differential BMI. Such an explanation would reinforce methodological rigour and provide a clear context on the relevance of these variables.
Lignes 113-115 : "The numerical variables were expressed as mean and Standard deviation..."
Comment: Mention whether statistical tests other than the t-test were considered for the comparisons.
Discussion
Lines 267-268: ‘The findings regarding family size and BMI further illustrate how dietary habits...’.
Comment: clarify whether this trend is consistent with other similar research.
The addition of a section dedicated to the strengths and weaknesses of the study would be beneficial in enhancing its transparency and credibility.
Conclusions
Lines 303-305: ‘Future research could further explore the long-term impacts...’.
Comment: Suggest specific methodologies for future studies (e.g. longitudinal cohorts, qualitative approaches).
Author Response
Reviewer 1
Introduction
Lines 41-42: ‘For example, fish has a simultaneous presence of vitamins, proteins, and omega-3.’
Comment: Clarify by mentioning specific examples of vitamins present in fish (for example, vitamin D or B12).
Authors answer: we have clarified the vitamins present in fish by providing specific examples (line 50).
The introduction could be enriched by adding clear hypotheses or research questions to give more structure to the article and guide the reader to the precise aims of the study
Authors answer: we have enriched the introduction by including explicit hypotheses and research questions to enhance the article’s clarity and focus (lines 102-109).
Research design
Comments: it would be interesting to mention more details on potential limitations, such as the self-selection bias of respondents to the questionnaire.
Authors answer: we thank the referee for the useful advice and have added the following sentence about possible limitations and future developments: “We are aware that the research design has a limit, due to the self-selection bias: in fact, each pregnant woman has the possibility to choose whether to adhere (or not) to the compilation of the questionnaire and, therefore, to become part of the sample subject to statistical analysis. A pregnant woman who, for various reasons, does not provide consent to participate in the survey represents an impoverishment of available information and, consequently, this involves a self-selection of the interviewees, which cannot be controlled by the researcher.”
Lines 97-99: ‘Specifically, we used the cohort of 114 pregnant women enrolled in 2013...’.
Comment: Explain briefly why these specific cohorts were chosen.
Authors answer: the reason why the two cohorts were chosen is due to the fact that the two cohorts of women agreed to fill out the same questionnaire. Therefore, the same variables were detected in the same form, in two different periods. In addition, both cohorts are composed of pregnant women who consume fresh fish and frozen fish.
Methods
The methods are detailed but lack justification for certain approaches, such as excluding women who do not eat fish.
Authors answer: we specify that the aim of our study is to evaluate the benefits related to the frequency of consumption of fresh and frozen fish in pregnant women, therefore it was considered appropriate to exclude women who do not consume any type of fish, as the latter do not provide any type of useful information to the survey.
Comments: adding information on the validation of the questionnaire used and the reasons behind the sampling chosen would strengthen this section.
Authors answer: the questionnaire is to be considered validated since it was administered first in 2013 (reference 21) and after 10 years in 2023 (reference 22) and allowed to collect useful information on the eating habits of pregnant women.
Authors answer: in Material and Method section, we added the following references: The sampling chosen guarantees representativeness as it allows to reach a small “photograph” of the pregnant female population that consumes fish; the chosen sample allows to reach conclusions that can be generalized to the reference population, thanks also to the used inferential procedures.
Lines 104-106: ‘Since the first cohort is composed of women who consume both fresh and frozen fish...’.
Comment: Justify why women who do not eat fish or who eat only fresh fish were excluded.
Authors answer: since the aim of our study is to evaluate the benefits related to the frequency of consumption of fresh and frozen fish in pregnant women, we considered appropriate to exclude women who do not consume any type of fish or who eat only fresh fish.
Lines 108-111: The methods section could be enriched by explaining why the following variables were considered in the statistical analysis: age, education level, occupation, family size, pre-pregnancy BMI, and differential BMI. Such an explanation would reinforce methodological rigour and provide a clear context on the relevance of these variables.
Authors answer: we added the following sentence: “These variables were detected in the two cohorts and compared with each other. The evaluation of each analyzed variable assumes a great influence: age, education level, occupation and family size represent the stratification variables; the pre-pregnancy BMI and differential BMI instead constitute indicators about the physical health status of the mothers”
Lignes 113-115: "The numerical variables were expressed as mean and Standard deviation..."
Comment: Mention whether statistical tests other than the t-test were considered for the comparisons.
Authors answer: as we reported in our manuscript, Student t-test is the only statistical method used to to compare the two cohorts of pregnant women with reference to age, pre-pregnancy BMI, and differential BMI.
Discussion
Lines 267-268: ‘The findings regarding family size and BMI further illustrate how dietary habits...’.
Comment: clarify whether this trend is consistent with other similar research.
Authors answer: we have revised the discussion to explicitly clarify that the observed trend is consistent with findings from other research (lines 320-333).
The addition of a section dedicated to the strengths and weaknesses of the study would be beneficial in enhancing its transparency and credibility.
Authors answer: we have reported the strengths and limitations at the end of the Discussion section (lines 337-349).
Conclusions
Lines 303-305: ‘Future research could further explore the long-term impacts...’.
Comment: Suggest specific methodologies for future studies (e.g. longitudinal cohorts, qualitative approaches).
Authors answer: we added the following sentence: “In future perspectives, longitudinal cohorts could be started to investigate other possible variables related to fish consumption during pregnancy and the possible benefits deriving from it.

Reviewer 2 Report
Comments and Suggestions for Authors
Dear Authors,
There are numerous studies on the health benefits of consuming fish, such as DHA and EPA, but there are relatively few papers that address how to encourage people to consume fish. The content of your study is highly intriguing and provides valuable insights.
Could you include data, if available, on how fish prices have changed between 2013 and 2023? If prices have decreased, it would naturally explain the increase in consumption.
If you have data on family structure, please consider adding it. Specifically, differences between households with only a father and children, a mother and children, or both parents and children, as well as whether both parents are working or not. These factors likely influence the amount of time available for cooking, which could have a significant impact on the choice to prepare fish dishes. Additionally, differences in the number of children in the household and the age of those children should also be taken into consideration.
If possible, please examine the influence of age groups on fish purchasing behavior.
The relationship between household income and fish prices, as well as the amount of time a household can dedicate to domestic work, likely has a significant impact on whether fish is prepared or not. I would appreciate it if you could explore these aspects further and consider how these findings might inform approaches to increase fish consumption in the future.
Sincerely,
Author Response
Reviewer 2
Introduction
The entire introduction is irrelevant to the topic of the manuscript
The references to COVID are completely unnecessary here
Re-write the introduction and focus on your main topic (look at the title)
Authors answer: we sincerely appreciate your valuable suggestion regarding the introduction. As per your recommendation, we have thoroughly revised this section to better align with the main topic of the manuscript, as highlighted by the title. The references to COVID have been removed, and the introduction has been rewritten to focus exclusively on the core subject of our work.
If you use quote sequences (e.g. line 46 "Line 46: "[5,6,7,8,9,10,11,12]") - change it as follows (for the given example [5-12])
Authors answer: thank you for your thoughtful suggestion regarding the formatting of quote sequences. As recommended, we have updated all such instances in the manuscript to use the more concise format
Material and methods:
"Specifically, we used the cohort of 114 pregnant women enrolled in 2013 [21] and the cohort of 248 women enrolled in 2023 [22] who filled out the same questionnaire administered to evaluate their eating habits during pregnancy." - explain to me if these are the results of previously published studies? Do you have the consent of all authors to reuse this data?
Authors answer: certainly, this paper examines the results of previous studies in the bibliography [21 and 22]. In addition, we specify that many of the authors of the aforementioned papers coincide with those of the present manuscript, thus guaranteeing the continuity of the research work. The remaining authors have given their consent to the reuse of the data.
I have no comments on the way the results are described and presented, or on the discussion.
Authors answer: thank you for your kind feedback. We truly appreciate it.
The conclusions are supported by the results and are logically connected with them.
Authors answer: thank you for your kind feedback. We truly appreciate it.
There is no indication of the weaknesses and limitations of this study.
Authors answer: we thank the referee for the useful advice and have added the following sentence about possible limitations and future developments: “We are aware that the research design has a limit, due to the self-selection bias: in fact, each pregnant woman has the possibility to choose whether to adhere (or not) to the compilation of the questionnaire and, therefore, to become part of the sample subject to statistical analysis. A pregnant woman who, for various reasons, does not provide consent to participate in the survey represents an impoverishment of available information and, consequently, this involves a self-selection of the interviewees, which cannot be controlled by the researcher.”
The formatting of the reference list is not compliant with MDPI guidelines - please correct it
References are quite old, increase the number of newer citations (from 2020-2024)
Authors answer: thank you for your suggestions. We have updated the reference list to comply with MDPI guidelines and added more recent citations (2020–2024).
The full survey form should be available to the reader - include this form in the additional materials or as a supplement (in English and in the original language, if it was other than English)
Authors answer: we have included the survey form.

Reviewer 3 Report
Comments and Suggestions for Authors
My comments:
1. Introduction -- general note
a. - the entire introduction is irrelevant to the topic of the manuscript
b. - the references to COVID are completely unnecessary here
c. - re-write the introduction and focus on your main topic (look at the title)
2. if you use quote sequences (e.g. line 46 "Line 46: "[5,6,7,8,9,10,11,12]") - change it as follows (for the given example [5-12])
3. Material and methods:
a. "Specifically, we used the cohort of 114 pregnant women enrolled in 2013 [21] and the cohort of 248 women enrolled in 2023 [22] who filled out the same questionnaire administered to evaluate their eating habits during pregnancy." - explain to me if these are the results of previously published studies? Do you have the consent of all authors to reuse this data?
4. I have no comments on the way the results are described and presented, or on the discussion.
5. The conclusions are supported by the results and are logically connected with them.
6. There is no indication of the weaknesses and limitations of this study.
7. The formatting of the reference list is not compliant with MDPI guidelines - please correct it
8. References are quite old, increase the number of newer citations (from 2020-2024)
9. the full survey form should be available to the reader - include this form in the additional materials or as a supplement (in English and in the original language, if it was other than English)
Author Response
Reviewer 3
Introduction
The entire introduction is irrelevant to the topic of the manuscript
The references to COVID are completely unnecessary here
Re-write the introduction and focus on your main topic (look at the title)
Authors answer: we sincerely appreciate your valuable suggestion regarding the introduction. As per your recommendation, we have thoroughly revised this section to better align with the main topic of the manuscript, as highlighted by the title. The references to COVID have been removed, and the introduction has been rewritten to focus exclusively on the core subject of our work.
If you use quote sequences (e.g. line 46 "Line 46: "[5,6,7,8,9,10,11,12]") - change it as follows (for the given example [5-12])
Authors answer: thank you for your thoughtful suggestion regarding the formatting of quote sequences. As recommended, we have updated all such instances in the manuscript to use the more concise format
Material and methods:
"Specifically, we used the cohort of 114 pregnant women enrolled in 2013 [21] and the cohort of 248 women enrolled in 2023 [22] who filled out the same questionnaire administered to evaluate their eating habits during pregnancy." - explain to me if these are the results of previously published studies? Do you have the consent of all authors to reuse this data?
Authors answer: certainly, this paper examines the results of previous studies in the bibliography [21 and 22]. In addition, we specify that many of the authors of the aforementioned papers coincide with those of the present manuscript, thus guaranteeing the continuity of the research work. The remaining authors have given their consent to the reuse of the data.
I have no comments on the way the results are described and presented, or on the discussion.
Authors answer: thank you for your kind feedback. We truly appreciate it.
The conclusions are supported by the results and are logically connected with them.
Authors answer: thank you for your kind feedback. We truly appreciate it.
There is no indication of the weaknesses and limitations of this study.
Authors answer: we thank the referee for the useful advice and have added the following sentence about possible limitations and future developments: “We are aware that the research design has a limit, due to the self-selection bias: in fact, each pregnant woman has the possibility to choose whether to adhere (or not) to the compilation of the questionnaire and, therefore, to become part of the sample subject to statistical analysis. A pregnant woman who, for various reasons, does not provide consent to participate in the survey represents an impoverishment of available information and, consequently, this involves a self-selection of the interviewees, which cannot be controlled by the researcher.”
The formatting of the reference list is not compliant with MDPI guidelines - please correct it
References are quite old, increase the number of newer citations (from 2020-2024)
Authors answer: thank you for your suggestions. We have updated the reference list to comply with MDPI guidelines and added more recent citations (2020–2024).
The full survey form should be available to the reader - include this form in the additional materials or as a supplement (in English and in the original language, if it was other than English)
Authors answer: we have included the survey form.
Round 2
Reviewer 3 Report
Comments and Suggestions for Authors
The references are still not formatted according to MDPI guidelines. I have no other comments.